# Time-dependent spatially varying graphical models, with application to brain fMRI data analysis

**Kristjan Greenewald**
Department of Statistics
Harvard University

**Seyoung Park**
Department of Biostatistics
Yale University

**Shuheng Zhou**
Department of Statistics
University of Michigan

**Alexander Giessing**
Department of Statistics
University of Michigan

## Abstract

In this work, we present an additive model for space-time data that splits the data into a temporally correlated component and a spatially correlated component. We model the spatially correlated portion using a time-varying Gaussian graphical model. Under assumptions on the smoothness of changes in covariance matrices, we derive strong single sample convergence results, confirming our ability to estimate meaningful graphical structures as they evolve over time. We apply our methodology to the discovery of time-varying spatial structures in human brain fMRI signals.

## 1   Introduction

Learning structured models of high-dimensional datasets from relatively few training samples is an important task in statistics and machine learning. Spatiotemporal data, in the form of $n$ variables evolving over $m$ time points, often fits this regime due to the high ($mn$) dimension and potential difficulty in obtaining independent samples. In this work, we develop a nonparametric framework for estimating time varying spatiotemporal graphical structure using an $\ell 1$ regularization method. The covariance of a spatiotemporal array $X = [x^1, \ldots, x^m] \in \mathbb{R}^{n \times m}$ is an $mn$ by $mn$ matrix

$$\Sigma = \text{Cov}\left[\text{vec}([x^1, \ldots, x^m])\right], \tag{1}$$

where $x^i \in \mathbb{R}^n$, $i = 1, \ldots, m$ denotes the $n$ variables or features of interest at the $i$th time point. Even for moderately large $m$ and $n$ the number of degrees of freedom $(mn(mn + 1)/2)$ in the covariance matrix can greatly exceed the number of training samples available for estimation. One way to handle this problem is to introduce structure and/or sparsity, thus reducing the number of parameters to be estimated. Spatiotemporal data is often highly structured, hence the design of estimators that model and exploit appropriate covariance structure can provide significant gains.

We aim to develop a nonparametric framework for estimating time varying graphical structure for matrix-variate distributions. Associated with each $x^i \in \mathbb{R}^n$ is its undirected graph $G(i)$. Under the assumption that the law $\mathcal{L}(x^i)$ of $x^i$ changes smoothly, Zhou et al. (2010) introduced a nonparametric method to estimate the graph sequence $G(1), G(2), \ldots$ assuming that the $x^i \sim \mathcal{N}(0, B(i/m))$ are independent, where $B(t)$ is a smooth function over $t \in [0, 1]$ and we have mapped the indices $i$ onto points $t = i/m$ on the interval $[0, 1]$. In this work, we are interested in the general time series model where the $x^i, i = 1, \ldots, m$ are dependent and the $B^{-1}(t)$ graphs change over time.

One way to introduce dependency into the $x^i$ is to study the following covariance structure. Let $A = (a_{ij}) \in \mathbb{R}^{m \times m}, B(t) = (b_{ij}(t)) \in \mathbb{R}^{n \times n}, t \in [0, 1]$ be symmetric positive definite covariance

matrices. Let $\mathrm{diag}(v)$, $v = (v_1, \ldots, v_m)$ be the diagonal matrix with elements $v_i$ along the diagonal. Consider the random matrix $X$ with row vectors $y^j$ corresponding to measurements at the $j$th spatial location, and columns $x^i$ corresponding to the $m$ measurements at times $i/m$, $i = 1, \ldots, m$:

$$\forall j = 1, \ldots, n, \ \ y^j \sim \mathcal{N}_m(0, A_j) \ \text{ where } \ A_j = A + \mathrm{diag}(b_{jj}(1), \ldots, b_{jj}(m)), \ \text{ and} \tag{2}$$

$$\forall i = 1, \ldots, m, \ \ x^i \sim \mathcal{N}_n(0, a_{ii}I + B(i/m)) \ \text{ where } B(t) \text{ changes smoothly over } t \in [0, 1]; \tag{3}$$

that is, the covariance of the column vectors $x^i$ corresponding to each time point changes smoothly with time (if $a_{ii}$ is a smooth function of $i$). This provides ultimate flexibility in parameterizing spatial correlations, for example, across different geographical scales through variograms (Cressie, 2015), each of which is allowed to change over seasons. Observe that while we have used the normal distribution here for simplicity, all our results hold for general subgaussian distributions.

The model (3) also allows modeling the dynamic gene regulatory and brain connectivity networks with topological (e.g., Erdős-Rényi random graph, small-world graph, or modular graphs) constraints via degree specifications as well as spatial constraints in the set of $\{B(t), t = 1, 2, \ldots\}$. When $A = \mathbf{0}$, we return to the case of Zhou et al. (2010) where there is no temporal correlation, i.e. $y^1, \ldots, y^n$ assumed to be independent.

We propose methodologies to study the model as constructed in (2) and (3). Building upon and extending techniques of Zhou et al. (2010) and Rudelson & Zhou (2017); Zhou (2014), we aim to design estimators to estimate graph sequence $G(1), G(2), \ldots$, where the temporal graph $H$ and spatial graphs $G(i)$ are determined by the zeros of $A^{-1}$ and $B(t)^{-1}$. Intuitively, the temporal correlation and spatial correlation are modeled as two additive processes. The covariance of $X$ is now

$$\mathrm{Cov}[\mathrm{vec}(X)] = \Sigma = A \otimes I_n + \sum\nolimits_{i=1}^{m} (e_i e_i^T) \otimes B(i/m) \tag{4}$$

where $e_i \in \mathbb{R}^m$, $\forall i$ are the $m$-dimensional standard basis vectors.

In the context of this model, we aim to develop a nonparametric method for estimating time varying graphical structure for matrix-variate normal distributions using an $\ell_1$ regularization method. We will show that, as long as the covariances change smoothly over time, we can estimate the spatial and temporal covariance matrices well in terms of predictive risk even when $n, m$ are both large. We will investigate the following theoretical properties: (a) consistency and rate of convergence in the operator and Frobenius norm of the covariance matrices and their inverse, (b) large deviation results for covariance matrices for simultaneously correlated and non-identically distributed observations, and (c) conditions that guarantee smoothness of the covariances.

Besides the model (4), another well-studied option for modeling spatio-temporal covariances $\Sigma$ is to introduce structure via the Kronecker product of smaller symmetric positive definite matrices, i.e. $\Sigma = A \otimes B$. The Kronecker product model, however, is restrictive when applied to general spatio-temporal covariances as it assumes the covariance is separable (disallowing such simple scenarios as the presence of additive noise), and does not allow for time varying spatial structure. When used to estimate covariances not following Kronecker product structure, many estimators will respond to the model mismatch by giving ill-conditioned estimates (Greenewald & Hero, 2015).

Human neuroscience data is a notable application where time-varying structure emerges. In neuroscience, one must take into account temporal correlations as well as spatial correlations, which reflect the connectivity formed by the neural pathways. It is conceivable that the brain connectivity graph will change over a sufficiently long period of measurements. For example, as a child learns to associate symbols in the environment, certain pathways within the brain are reinforced. When they begin to associate images with words, the correlation between a particular sound like Mommy and the sight of a face becomes stronger and forms a well worn pathway. On the other hand, long term non-use of connections between sensory and motor neurons can result in a loss of the pathway.

## 1.1 Datasets and Related Work

Estimating graphical models (connectomes) in fMRI data using sparse inverse covariance techniques has enjoyed wide application (Huang et al., 2010; Varoquaux et al., 2010; Narayan et al., 2015; Kim et al., 2015). However, recent research has only now begun exploring observed phenomena such as temporal correlations and additive temporally correlated noise (Chen et al., 2015; Arbabshirani et al., 2014; Kim et al., 2015; Qiu et al., 2016), and time-varying dynamics and graphical models (connectomes) (Calhoun et al., 2014; Liu & Duyn, 2013; Chang & Glover, 2010; Chen et al., 2015).

We consider the ADHD-200 fMRI dataset (Biswal et al., 2010), and study resting state fMRIs for a variety of healthy patients in the dataset at different stages of development. Using our methods, we are able to directly estimate age-varying graphical models across brain regions, chronicling the development of brain structure throughout childhood.

Several models have emerged to generalize the Kronecker product model to allow it to model more realistic covariances while still maintaining many of the gains associated with Kronecker structure. Kronecker PCA, discussed in Tsiligkaridis & Hero (2013), approximates the covariance matrix using a sum of Kronecker products. An algorithm (Permuted Rank-penalized Least Squares (PRLS)) for fitting the KronPCA model to a measured sample covariance matrix was introduced in (Tsiligkaridis & Hero, 2013) and was shown to have strong high dimensional MSE performance guarantees. From a modeling perspective, the strengths of Kronecker PCA lie in its ability to handle "near separable" covariances and a variety of time-varying effects. While the Kronecker PCA model is very general, so far incorporation of sparsity in the inverse covariance has not been forthcoming. This motivates our introduction of the sparse model (4), which we demonstrate experimentally in Section 10 of the supplement to enjoy better statistical convergence.

Carvalho et al. (2007) proposed a Bayesian additive time-varying graphical model, where the spatially-correlated noise term is a parameter of the driving noise covariance in a temporal dynamical model. Unlike our method, they did not estimate the temporal correlation, instead requiring the dynamical model to be pre-set. Our proposed method has wholly independent spatial and temporal models, directly estimating an inverse covariance graphical model for the temporal relationships of the data. This allows for a much richer temporal model and increases its applicability.

In the context of fMRI, the work of Qiu et al. (2016) used a similar kernel-weighted estimator for the spatial covariance, however they modeled the temporal covariance with a simple AR-1 model which they did not estimate, and their estimator did not attempt to remove. Similarly, Monti et al. (2014) used a smoothed kernel estimator for $B^{-1}(t)$ with a penalty to further promote smoothness, but did not model the temporal correlations. Our additive model allows the direct estimation of the temporal behavior, revealing a richer structure than a simple AR-1, and allowing for effective denoising of the data, and hence better estimation of the spatial graph structures.

## 2 The model and method

Let the elements of $A \succ 0$ and $B(t)$ be denoted as $[A]_{ij} := a_{ij}$ and $[B(t)]_{ij} := b_{ij}(t), t \in [0, 1]$. Similar to the setting in (Zhou et al., 2010), we assume that $b_{ij}(t)$ is a smooth function of time $t$ for all $i, j$, and assume that $B^{-1}(t)$ is sparse. Furthermore, we suppose that $m \gg n$, corresponding to there being more time points than spatial variables. For a random variable $Y$, the subgaussian norm of $Y$, $\|Y\|_{\psi_2}$, is defined via $\|Y\|_{\psi_2} = \sup_{p \geq 1} p^{-1/2} (E|Y|^p)^{1/p}$. Note that if $E[Y] = 0$, we also have $E[\exp(tY)] \leq \exp(Ct^2 \|Y\|_{\psi_2}^2) \quad \forall t \in \mathbb{R}$. Define an $n \times m$ random matrix $Z$ with independent, zero mean entries $Z_{ij}$ satisfying $E[Z_{ij}^2] = 1$ and having subgaussian norm $\|Z_{ij}\|_{\psi_2} \leq K$. Matrices $Z_1, Z_2$ denote independent copies of $Z$. We now write an additive generative model for subgaussian data $X \in \mathbb{R}^{n \times m}$ having covariance given in (4). Let

$$X = Z_1 A^{1/2} + Z_B \tag{5}$$

where $Z_B = [B(1/m)^{1/2} Z_2 e_1, \ldots, B(i/m)^{1/2} Z_2 e_i, \ldots, B(1)^{1/2} Z_2 e_m]$, and $e_i \in \mathbb{R}^m$, $\forall i$ are the $m$-dimensional standard basis vectors. Then the covariance

$$\Sigma = \mathrm{Cov}[\mathrm{vec}(X)] = \mathrm{Cov}[\mathrm{vec}(Z_1 A^{1/2})] + \mathrm{Cov}[\mathrm{vec}(Z_B)]$$
$$= \mathrm{Cov}[\mathrm{vec}(Z_1 A^{1/2})] + \sum_{i=1}^{m} (e_i e_i^T) \otimes \mathrm{Cov}[B(i/m)^{1/2} Z_2 e_i]$$
$$= A \otimes I_n + \sum_{i=1}^{m} (e_i e_i^T) \otimes B(i/m).$$

Thus (5) is a generative model for data following the covariance model (4).

### 2.1 Estimators

As in Rudelson & Zhou (2017), we can exploit the large-$m$ convergence of $Z_1 A Z_1^T$ to $\mathrm{tr}(A)I$ to project out the $A$ part and create an estimator for the $B$ covariances. As $B(t)$ is time-varying, we use a weighted average across time to create local estimators of spatial covariance matrix $B(t)$.

It is often assumed that knowledge of the trace of one of the factors is available a priori. For example, the spatial signal variance may be known and time invariant, corresponding to $\mathrm{tr}(B(t))$ being known. Alternatively, the temporal component variance may be constant and known, corresponding to $\mathrm{tr}(A)$ being known. In our analysis below, we suppose that $\mathrm{tr}(A)$ is known or otherwise estimated (similar results hold when $\mathrm{tr}(B(t))$ is known). For simplicity in stating the trace estimators, in what follows we suppose that $\mathrm{tr}(B(t)) = \mathrm{tr}(B)$ is constant, and without loss of generality that the data has been normalized such that diagonal elements $A_{ii}$ are constant over $i$.

As $B(t)$ is smoothly varying over time, the estimate at time $t_0$ should depend strongly on the time samples close to $t_0$, and less on the samples further from $t_0$. For any time of interest $t_0$, we thus construct a weighted estimator using a weight vector $w_i(t_0)$ such that $\sum_{t=1}^{m} w_t(t_0) = 1$. Our weighted, unstructured sample-based estimator for $B(t_0)$ is then given by

$$\widehat{S}_m(t_0) := \sum_{i=1}^{m} w_i(t_0)\left(x_i x_i^T - \frac{\mathrm{tr}(A)}{m}I_n\right), \quad \text{where} \quad w_i(t_0) = \frac{1}{mh}K\left(\frac{i/m - t_0}{h}\right), \quad (6)$$

and we have considered the class of weight vectors $w_i(t_0)$ arising from a symmetric nonnegative kernel function $K$ with compact support $[0, 1]$ and bandwidth determined by parameter $h$. A list of minor regularity assumptions on $K$ are listed in the supplement. For kernels such as the Gaussian kernel, this $w_t(t_0)$ will result in samples close to $t_0$ being highly weighted, with the "weight decay" away from $t_0$ scaling with the bandwidth $h$. A wide bandwidth will be appropriate for slowly-varying $B(t)$, and a narrow bandwidth for quickly-varying $B(t)$.

To enforce sparsity in the estimator for $B^{-1}(t_0)$, we substitute $\widehat{S}_m(t_0)$ into the widely-used GLasso objective function, resulting in a penalized estimator for $B(t_0)$ with regularization parameter $\lambda_m$

$$\widehat{B}_\lambda(t_0) := \arg\min_{B_\lambda \succ 0} \mathrm{tr}\left(B_\lambda^{-1}\widehat{S}_m(t_0)\right) + \log|B_\lambda| + \lambda_m|B_\lambda^{-1}|_1. \quad (7)$$

For a matrix $B$, we let $|B|_1 := \sum_{ij}|B_{ij}|$. Increasing the parameter $\lambda_m$ gives an increasingly sparse $\widehat{B}_\lambda^{-1}(t_0)$. Having formed an estimator for $B$, we can now form a similar estimator for $A$. Under the constant-trace assumption, we construct an estimator for $\mathrm{tr}(B)$

$$\hat{\mathrm{tr}}(B) = \sum_{i=1}^{m} w_i\|X_i\|_2^2 - \frac{n}{m}\mathrm{tr}(A), \text{ with } w_i = \frac{1}{m}. \quad (8)$$

For a time-varying trace $\mathrm{tr}(B(t))$, use the time-averaged kernel

$$\hat{\mathrm{tr}}(B(t_0)) = \sum_{i=1}^{m} w_i(t_0)\|X_i\|_2^2 - \frac{n}{m}\mathrm{tr}(A), \text{ with } w_i(t_0) = \frac{1}{mh}K\left(\frac{i/m - t_0}{h}\right). \quad (9)$$

In the future we will derive rates for the time varying case by choosing an optimal $h$. These estimators allow us to construct a sample covariance matrix for $A$:

$$\tilde{A} = \frac{1}{n}X^TX - \frac{1}{n}\mathrm{diag}\{\hat{\mathrm{tr}}(B(1/m)), \dots, \hat{\mathrm{tr}}(B(1))\}. \quad (10)$$

We (similarly to $B(t)$) apply the GLasso approach to $\tilde{A}$. Note that with $m > n$, $\tilde{A}$ has negative eigenvalues since $\lambda_{\min}\left(\frac{1}{n}X^TX\right) = 0$. We obtain a positive semidefinite matrix $\tilde{A}_+$ as:

$$\tilde{A}_+ = \arg\min_{A \succeq 0}\|\tilde{A} - A\|_{\max}. \quad (11)$$

We use alternating direction method of multipliers (ADMM) to solve (11) as in Boyd et al. (2011), and prove that this retains a tight elementwise error bound. Note that while we chose this method of obtaining a positive semidefinite $\tilde{A}_+$ for its simplicity, there may exist other possible projections, the exact method is not critical to our overall Kronecker sum approach. In fact, if the GLasso is not used, it is not necessary to do the projection (11), as the elementwise bounds also hold for $\tilde{A}$.

We provide a regularized estimator for the correlation matrices $\rho(A) = \mathrm{diag}(A)^{-1/2}A\,\mathrm{diag}(A)^{-1/2}$ using the positive semidefinite $\tilde{A}_+$ as the initial input to the GLasso problem

$$\hat{\rho}_\lambda(A) = \mathrm{argmin}_{A_\rho \succ 0}\,\mathrm{tr}(A_\rho^{-1}\rho(\tilde{A}_+)) + \log|A_\rho| + \lambda_n|A_\rho^{-1}|_{1,\mathrm{off}}, \quad (12)$$

where $\lambda_n > 0$ is a regularization parameter and $|\cdot|_{1,\mathrm{off}}$ is the L1 norm on the offdiagonal.

Form the estimate for $A$ as $\frac{\mathrm{tr}(A)}{m}\hat{\rho}_\lambda(A)$. Observe that our method has three tuning parameters, two if $\mathrm{tr}(A)$ is known or can be estimated. If $\mathrm{tr}(A)$ is not known, we present several methods to choose it in Section 7.1 in the supplement. Once $\mathrm{tr}(A)$ is chosen, the estimators (7) and (12) for $A$ and $B(t)$ respectively do not depend on each other, allowing $\lambda_m$ and $\lambda_n$ to be tuned independently.

# 3 Statistical convergence

We first bound the estimation error for the time-varying $B(t)$. Since $\hat{B}(t)$ is based on a kernel-smoothed sample covariance, $\hat{B}(t)$ is a biased estimator, with the bias depending on the kernel width and the smoothness of $B(t)$. In Section 12.1 of the supplement, we derive the bias and variance of $\hat{S}_m(t_0)$, using arguments from kernel smoothing and subgaussian concentration respectively.

In the following results, we assume that the eigenvalues of the matrices $A$ and $B(t)$ are bounded:

**Assumption 1**: There exist positive constants $c_A, c_B$ such that $\frac{1}{c_A} \leq \lambda_{\min}(A) \leq \lambda_{\max}(A) \leq c_A$ and $\frac{1}{c_B} \leq \lambda_{\min}(B(t)) \leq \lambda_{\max}(B(t)) \leq c_B$ for all $t$.

**Assumption 2**: $B(t)$ has entries with bounded second derivatives on $[0, 1]$.

Putting the bounds on the bias and variance together and optimizing the rate of $h$, we obtain the following, which we prove in the supplementary material.

**Theorem 1.** *Suppose that the above Assumption holds, the entries $B_{ij}(t)$ of $B(t)$ have bounded second derivatives for all $i, j$, and $t \in [0, 1]$, $s_b + n = o((m/\log m)^{2/3})$, and that $h \asymp (m^{-1}\log m)^{1/3}$. Then with probability at least $1 - \frac{c''}{m^{8/3}}$, $\hat{S}_m(t_0)$ is positive definite and for some $C$*

$$\max_{ij}|\hat{S}_m(t_0, i, j) - B(t_0, i, j)| \leq C\left(m^{-1}\log m\right)^{1/3}.$$

This result confirms that the $mh$ temporal samples selected by the kernel act as replicates for estimating $B(t)$. We can now substitute this elementwise bound on $\hat{S}_m(t_0)$ into the GLasso proof, obtaining the following theorem which demonstrates that $\hat{B}(t)$ successfully exploits sparsity in $B^{-1}(t)$.

**Theorem 2.** *Suppose the conditions of Theorem 1 and that $B^{-1}(t)$ has at most $s_b$ nonzero off-diagonal elements for all $t$. If $\lambda_m \sim \sqrt{\frac{\log m}{m^{2/3}}}$, then the GLasso estimator (7) satisfies*

$$\|\hat{B}(t_0) - B(t_0)\|_F = O_p\left(\sqrt{\frac{(s_b + n)\log m}{m^{2/3}}}\right), \|\hat{B}^{-1}(t_0) - B^{-1}(t_0)\|_F = O\left(\sqrt{\frac{(s_b + n)\log m}{m^{2/3}}}\right)$$

Observe that this single-sample bound converges whenever the $A$ part dimensionality $m$ grows. The proof follows from the concentration bound in Theorem 1 using the argument in Zhou et al. (2010), Zhou et al. (2011), and Rothman et al. (2008). Note that $\lambda_m$ goes to zero as $m$ increases, in accordance with the standard bias/variance tradeoff.

We now turn to the estimator for the $A$ part. As it does not involve kernel smoothing, we simply need to bound the variance. We have the following bound on the error of $\tilde{A}$:

**Theorem 3.** *Suppose the above Assumption holds. Then*

$$\max_{ij}|\tilde{A}_{ij} - A_{ij}| \leq C(c_A + c_B)\sqrt{n^{-1}\log m}$$

*with probability $1 - \frac{c}{m^4}$ for some constants $C, c > 0$.*

Recall that we have assumed that $m > n$, so the probability converges to 1 with increasing $m$ or $n$. While $\tilde{A}$ is not positive definite, the triangle inequality implies a bound on the positive definite projection $\tilde{A}_+$ (11):

$$\|\tilde{A}_+ - A\|_{\max} \leq \|\tilde{A}_+ - \tilde{A}\|_{\max} + \|\tilde{A} - A\|_{\max} \leq 2\|\tilde{A} - A\|_{\max} = O_p\left(\sqrt{n^{-1}\log m}\right). \quad (13)$$

Thus, similarly to the earlier result for $B(t)$, the estimator (12) formed by substituting the positive semidefinite $\rho(\tilde{A}_+)$ into the GLasso objective enjoys the following error bound (Zhou et al., 2011).

**Theorem 4.** *Suppose the conditions of Theorem 3 and that $A^{-1}$ has at most $s_a = o(n/\log m)$ nonzero off-diagonal elements. If $\lambda_n \sim \sqrt{\frac{\log m}{n}}$, then the GLasso estimator (12) satisfies*

$$\|\hat{A} - A\|_F = O_p\left(\sqrt{\frac{s_a \log m}{n}}\right), \quad \|\hat{A}^{-1} - A^{-1}\|_F = O_p\left(\sqrt{\frac{s_a \log m}{n}}\right).$$

Observe that this single-sample bound converges whenever the $B(t)$ dimensionality $n$ grows since the sparsity $s_a = o(n/\log m)$. For relaxation of this stringent sparsity assumption, one can use other assumptions, see for example Theorem 3.3 in Zhou (2014).

## 4 Simulation study

We generated a time varying sequence of spatial covariances $B(t_i) = \Theta(t_i)^{-1}$ according to the method of Zhou et al. (2010), which follows a type of Erdos-Renyi random graph model. Initially we set $\Theta(0) = 0.25 I_{n \times n}$, where $n = 100$. Then, we randomly select $k$ edges and update $\Theta(t)$ as follows: for each new edge $(i, j)$, a weight $a > 0$ is chosen uniformly at random from $[0.1, 0.3]$; we subtract $a$ from $\Theta_{ij}, \Theta_{ji}$, and increase $\Theta_{ii}, \Theta_{jj}$ by $a$. This keeps $B(t)$ positive definite. When we later delete an existing edge from the graph, we reverse the above procedure.

We consider $t \in [0, 1]$, changing the graph structure at the points $t_i = i/5$ as follows. At each $t_i$, five existing edges are deleted, and five new edges are added. For each of the five new edges, a target weight is chosen. Linear interpolation of the edge weights between the $t_i$ is used to smoothly add the new edges and gradually delete the ones to be deleted. Thus, almost always, there are 105 edges in the graph and 10 edges have weights that are varying smoothly (Figure 1).

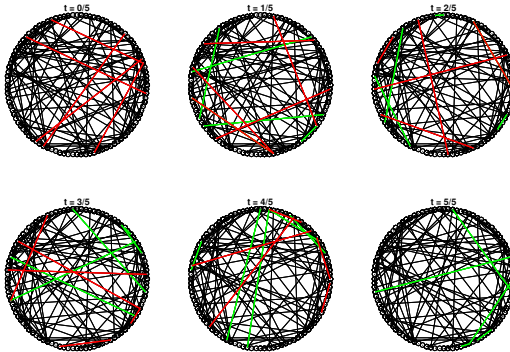

Figure 1: Example sequence of Erdos-Renyi $B^{-1}(t) = \Theta(t)$ graphs. At each time point, the 100 edges connecting $n = 100$ nodes are shown. Changes are indicated by red and green edges: red edges indicate edges that will be deleted in the next increment and green indicates new edges.

In the first set of experiments we consider $B(t)$ generated from the ER time-varying graph procedure and $A$ an AR-1 covariance with parameter $\rho$. The magnitudes of the two factors are balanced. We set $n = 100$ and vary $m$ from 200 to 2400. For each $n, m$ pair, we vary the $B(t)$ regularization parameter $\lambda$, estimating every $B(t)$, $t = 1/m, \ldots, 1$ for each. We evaluate performance using the mean relative Frobenius $B(t)$ estimation error ($\|\hat{B}(t) - B(t)\|_F / \|B(t)\|_F$), the mean relative L2 estimation error ($\|\hat{B}(t) - B(t)\|_2 / \|B(t)\|_2$), and the Matthews correlation coefficient (MCC).

The MCC quantifies edge support estimation performance, and is defined as follows. Let the number of true positive edge detections be TP, true negatives TN, false positives FP, and false negatives FN. The Matthews correlation coefficient is defined as $\mathrm{MCC} = \frac{\mathrm{TP \cdot TN - FP \cdot FN}}{\sqrt{(\mathrm{TP+FP})(\mathrm{TP+FN})(\mathrm{TN+FP})(\mathrm{TN+FN})}}$. Increasing values of MCC imply better edge estimation performance, with $\mathrm{MCC} = 0$ implying complete failure and $\mathrm{MCC} = 1$ implying perfect edge set estimation.

Results are shown in Figure 2, for $\rho = .5$ and 50 edges in $B$, $\rho = .5$ and 100 edges in $B$, and $\rho = .95$ and 100 edges in $B$. As predicted by the theory, increasing $m$ improves performance and increasing $\rho$ decreases performance. Increasing the number of edges in $B$ changes the optimal $\lambda$, as expected. Figure 3 shows performance results for the penalized estimator $\hat{A}$ using MCC, Frobenius error, and L2 error, where $A$ follows an AR(1) model with $\rho = 0.5$ and $B$ follows a random ER model. Note the MCC, Frobenius, spectral norm errors are improved with larger $n$. In the supplement (Section 11), we repeat these experiments, using an alternate random graph topologies, with similar results.

## 5 fMRI Application

The ADHD-200 resting-state fMRI dataset (Biswal et al., 2010) was collected from 973 subjects, 197 of which were diagnosed with ADHD types 1, 2, or 3. The fMRI images have varying numbers

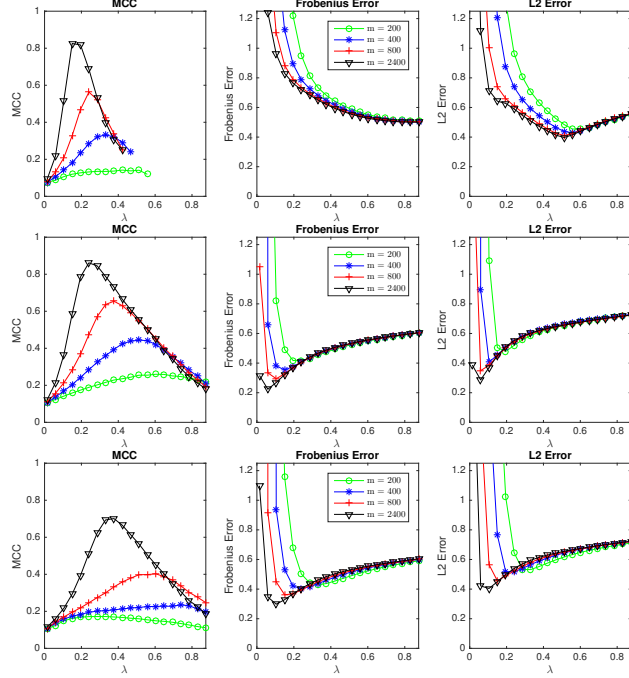

Figure 2: MCC, Frobenius, and L2 norm error curves for $B$ a random ER graph and $n = 100$. Top: $A$ is AR covariance with $\rho = .5$ and 50 edges in $B$, Middle: $A$ is AR(1) covariance with $\rho = .5$ and $B$ having 100 edges, Bottom: AR covariance with $\rho = .95$ and 100 edges in $B$.

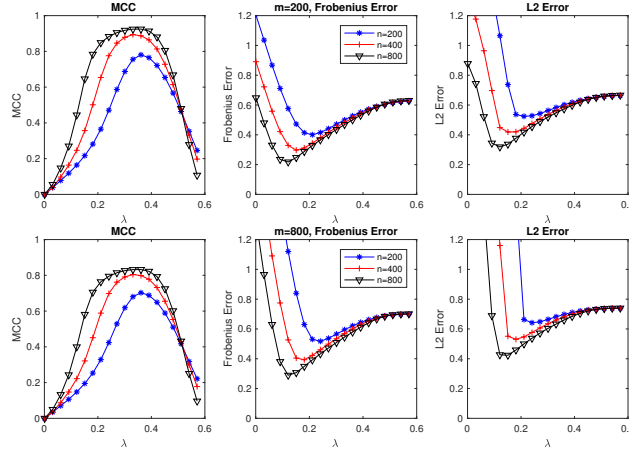

Figure 3: MCC, Frobenius, and L2 norm error curves for $A$ a AR(1) with $\rho = 0.5$ when $B$ is a random ER graph. From top to bottom: $m = 200$ and $m = 800$.

of voxels which we divide into 90 regions of interest for graphical model analysis (Wehbe et al., 2014), and between 76 and 276 images exist for each subject. Provided covariates for the subjects include age, gender, handedness, and IQ. Previous works such as (Qiu et al., 2016) used this dataset to establish that the brain network density increases with age, corresponding to brain development as subjects mature. We revisit this problem using our additive approach. Our additive model allows the direct estimation of the temporal behavior, revealing a richer structure than a simple AR-1, and allowing for effectively a denoising of the data, and better estimation of the spatial graph structure.

We estimate the temporal $A$ covariances for each subject using the voxels contained in the regions of interest, with example results shown in Figure 5 in the supplement. We choose $\tau_B$ as the lower limit of the eigenvalues of $\frac{1}{n}X^T X$, as in the high sample regime it is an upper bound on $\tau_B$.

We then estimate the brain connectivity network at a range of ages from 8 to 18, using both our proposed method and the method of Monti et al. (2014), as it is an optimally-penalized version of the estimator in Qiu et al. (2016). We use a Gaussian kernel with bandwidth $h$, and estimate the graphs using a variety of values of $\lambda$ and $h$. Subjects with fewer than 120 time samples were eliminated, and those with more were truncated to 120 to reduce bias towards longer scans. The number of edges in the estimated graphs are shown in Figure 4. Note the consistent increase in network density with age, becoming more smooth with increasing $h$.

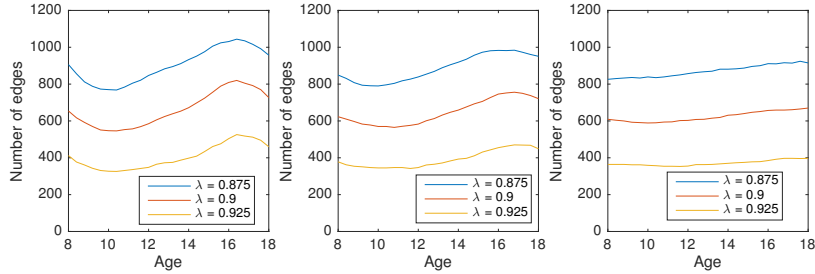

(a) Non-additive method of Monti et al. (2014) (optimally penalized version of Qiu et al. (2016)).

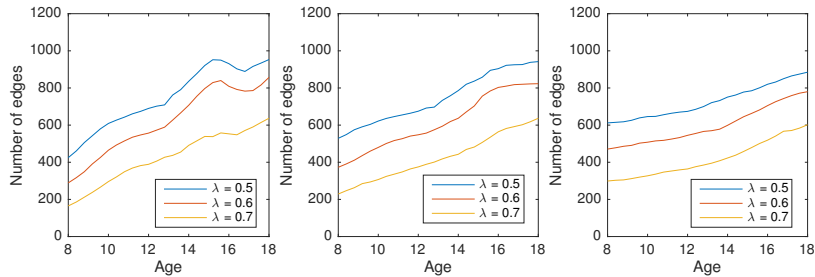

(b) Our proposed additive method, allowing for denoising of the time-correlated data.

Figure 4: Number of edges in the estimated $B^{-1}(t)$ graphical models across 90 brain regions as a function of age. Shown are results using three different values of the regularization parameter $\lambda$, and from left to right the kernel bandwidth parameter used is $h = 1.5, 2$, and 3. Note the consistently increasing edge density in our estimate, corresponding to predictions of increased brain connectivity as the brain develops, leveling off in the late teenage years. Compare this to the method of Monti et al. (2014), which successfully detects the trend in the years 11-14, but fails for other ages.

# 6   Conclusion

In this work, we presented estimators for time-varying graphical models in the presence of time-correlated signals and noise. We revealed a bias-variance tradeoff scaling with the underlying rate of change, and proved strong single sample convergence results in high dimensions. We applied our methodology to an fMRI dataset, discovering meaningful temporal changes in functional connectivity, consistent with scientifically expected childhood growth and development.

**Acknowledgement**

This work was supported in part by NSF under Grant DMS-1316731, Elizabeth Caroline Crosby Research Award from the Advance Program at the University of Michigan, and by AFOSR grant FA9550-13-1-0043.

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
