[Supplementary Material]

# Supplement to Time-dependent spatially varying graphical models, with application to brain fMRI data analysis

**Kristjan Greenewald**
Department of Statistics
Harvard University

**Seyoung Park**
Department of Biostatistics
Yale University

**Shuheng Zhou**
Department of Statistics
University of Michigan

**Alexander Giessing**
Department of Statistics
University of Michigan

## 7  Estimating covariance $A$ for fMRI data

We estimate the temporal covariance $A$ for each subject using the voxels contained in the regions of interest. The results for several example subjects are shown in Figure 5, along with the corresponding sample covariance $\frac{1}{n}X^T X$ and its eigenvalues. We choose $\tau_B$ as the lower "asymptote" of the eigenvalues of $\frac{1}{n}X^T X$, as in the high sample regime it is an upper bound on $\tau_B$.

The sample covariance matrix $\frac{1}{n}X^T X$ is significantly diagonally dominant, supporting our subtraction of a $\tau_B = \operatorname{tr}(B)/n$ scaled identity matrix. Note the evident sparsity in the inverse. Many of the sparsity patterns indicate local AR-type behavior as assumed in (Qiu et al., 2016), but this pattern is not stationary, and in fact tends to group in blocks. Hence, the assumptions in (Qiu et al., 2016) do not fully capture the richness of the data.

### 7.1  Choosing $\operatorname{tr}(A)$

Due to the nonidentifiability of $\operatorname{tr}(A)$ and $\operatorname{tr}(B)$ when a single copy of the data is observed, we assume that the $\operatorname{tr}(A)$ parameter is either known (Rudelson & Zhou, 2017), or chosen as a tuning parameter. If $A$ corresponds to a "signal" component in a physical system, then as the mean signal strength $\operatorname{tr}(A)/m$ may be known by design or may be estimated directly by running an experiment with the "noise" component involving the covariance parameter $B(t)$ to be set at 0 for a certain period of time.

The first step of relaxation is by assuming only one entry of $\operatorname{diag}(A)$, for example, $a_{11}$ is known. This is feasible when we assume we can observe for a short period of time only $X_0 := Z_1 A^{1/2}$ so as to obtain the knowledge of a single element in $\operatorname{diag}(A)$. It is our conjecture that there is an interesting tradeoff between the number of covariates in $X_0$ we are allowed to observe with no measurement errors and the rate of convergence we can obtain for estimating $A$ and $B$. Moreover, even when no covariate $X_0$ is observed directly, we can rely on the recent progress on high dimensional regression and signal reconstruction to help establish theoretical limits on recovering $\operatorname{tr}(A)$ and $\operatorname{tr}(B(t))$, when replicates are available. For example, if a second sample of either $Z_B$ or $X_0 = Z_1 A^{1/2}$ (cf. Equation (5)) is available, the corresponding $\operatorname{tr}(B(t))$ or $\operatorname{tr}(A)$ can be estimated directly without needing to specify any of $\operatorname{tr}(A)$ or $\operatorname{tr}(B(t))$. We leave these as future directions.

We now use two experimental settings to illustrate how robust the estimation procedure for covariance $A$ is with respect to the misspecification of $\operatorname{tr}(A)$. We consider the case in which $m = 400$, $n \in \{200, 400\}$, $\tau_A = \operatorname{tr}(A)/m = 1$, $\tau_B = 0.5$, and $A$ is generated using the AR(1) or Star-Block,

Figure 5: Estimated $A$ covariance for three example subjects. Left: $A$ part sample covariance $\frac{1}{n}X^T X$. Center: Eigenvalues of $\frac{1}{n}X^T X$, showing estimate of $\tau_B$ factor. Right: Estimated $A^{-1}$ graphical model. Note the sparsity in the inverse, and that the eigenvalue spectra are consistent with the additive model.

while $B$ follows random ER graph. We estimate the inverse of $A$ using $\hat{\tau}_A \in \{0.4, 0.5, \cdots, 1.4\}$ and $\lambda_A \in (0, 0.7)$, and use MCC to measure the performance of edge selection.

As shown in Figures 6-7, we observe that when the topology is sophisticated (e.g., Star-Block) and the misspecification error of $\hat{\tau}_A$ does not approach 0, although joint tuning of $\lambda$ and $\hat{\tau}_A$ can not resolve the edge selection inconsistency, as expected, although it appears to give certain improvements.

When the topology is relatively simple, e.g., like the chain graph correspondong to the AR(1) model, the edge selection performance is robust to the misspecification of $\hat{\tau}_A$; in this case, joint tuning of $\lambda$ and $\mathrm{tr}(A)$ is not even necessary, as illustrated in Figures 6-7).

# 8    Additional analysis of ADHD-200 data

In addition to the plots showing the relationship between brain connectivity and age in the main text, in this section we reproduce those plots using only healthy subjects and only using ADHD subjects. The results are shown in Figure 8.

Observe that the leftmost plots (those with the narrowest kernel) are quite rough. This is caused by the reduction in the number of available subjects as compared to the plots in the main text that used all the subjects, increasing the effect of the noise. Shown in Figure 9 are the age histograms for all patients, healthy patients only, and ADHD patients. Note the nonsmooth age ranges in the plots of Figure 8 correspond to regions with fewer available subjects, as expected.

While the sample size is such that we cannot make definitive conclusions on the exact form of the differences between healthy and ADHD brain development, the fact that we observe significant differences is not surprising given the nature of ADHD and its effects on childhood development. In particular, note the lower rate of development among teenage ADHD subjects as opposed to healthy

Figure 6: The figures are MCC of the estimators $\hat{A}$ with respect to the true $A$ as a function of $\lambda$ for each $\hat{\mathrm{tr}}(A) \in \{0.6, \cdots, 1.4\}$; $m = 400$ and $n = 200$; From left to right: $A$ is AR(1) and Star-Block model.

teenage subjects. We hypothesize that this corresponds to the common observation that ADHD patients face additional developmental challenges in the teenage years.

## 9 Technical assumptions

### 9.1 Assumptions

In this section, we repeat the assumptions that we required in the main text.

**Assumption A1** There exists a positive constant $c_A$ such that

$$\frac{1}{c_A} \leq \lambda_{\min}(A) \leq \lambda_{\max}(A) \leq c_A.$$

**Assumption A2** The diagonal elements $A_{ii}$ are constant across all $i$, and the trace $\mathrm{tr}(B(t))$ is constant over time.

**Assumption A3** $A^{-1}$ has at most $s_a = o(n/\log m)$ nonzero off-diagonal elements.

**Assumption B1** $B(t)$ is a symmetric and positive definite matrix for all $t$, and its entries have bounded second derivatives on $[0, 1]$.

**Assumption B2** There exists a positive constant $c_B$ such that for all $t \in [0, 1]$

$$1/c_B \leq \lambda_{\min}(B(t)) \leq \lambda_{\max}(B(t)) \leq c_B.$$

**Assumption B3** $B(t)^{-1}$ has at most $s_b + n = o((m/\log m)^{2/3})$ nonzero off-diagonal elements.

Note that two conditions in Assumption A2 can be relaxed, we include them only to make statement of the estimators easier. In the nonnormalized cases the appropriate modifications are easy to derive.

We make the following assumptions on the smooth kernel function $K(\cdot)$ used to create the $\widehat{S}_m(t)$:

**Assumption K1** $K(\cdot)$ is non-negative, symmetric, twice differentiable, and has compact support $[-1, 1]$.

Figure 7: The figures are MCC of the estimators $\hat{A}$ with respect to the true $A$ as a function of $\lambda$ for each $\hat{\text{tr}}(A) \in \{0.6, \cdots, 1.4\}$; $m = 400$ and $n = 400$; From left to right: $A$ is AR(1) and Star-Block model.

**Assumption K2** $\int K(u)du = 1$.

**Assumption K3** $\int u^2 K(u)du < \infty$.

**Assumption K4** $\sup_{u \in [-1,1]} K(u) \le K_1$.

**Assumption K5** $\sup_{u \in [-1,1]} K''(u/h) = O(h^{-4})$.

These are satisfied for most common smooth kernel functions, including the Gaussian kernel (Zhou et al., 2010).

## 10 Comparison of our method and Kronecker PCA

In this section, we compare our time-varying Kronecker sum model (4) to the sum of Kronecker products (KronPCA) model of Tsiligkaridis & Hero (2013); Greenewald & Hero (2015)

$$\Sigma = \sum_{i=1}^{r} A_i \otimes B_i.$$

Both our method and KronPCA are a sum of Kronecker products, KronPCA is a more general model, but our method exploits sparsity while KronPCA cannot. Consider data generated from a time-varying Kronecker sum model (4), where $A^{-1} \in \mathbf{R}^{60 \times 60}$ is a random ER graph and $B(t)^{-1} \in \mathbf{R}^{20 \times 20}$ is a time-varying random ER graph as in Section 4 in the main text. The sizes of $A, B$ where chosen to be relatively small since the computational complexity of KronPCA is $O(\min(m^6, n^6))$ (compared to the $O(m^3 + n^3)$ complexity of our method). Figure 10 shows Frobenius norm results for our L1-penalized method, KronPCA, and for comparison, the baseline sample covariance. Note that due to the sparsity of the true model, our method performs significantly better than KronPCA, especially when the number of available replicates is small. Since in a time-varying setting the number of replicates is small or even 1, this is a significant advantage. Additionally, our method provides interpretable graph estimates, while the factors $A_i, B_i$ of KronPCA are nonsparse and not interpretable Tsiligkaridis & Hero (2013); Greenewald & Hero (2015).

(a) Our proposed additive method, trained on healthy subjects only.

(b) Our proposed additive method, trained on ADHD subjects only.

Figure 8: Number of edges in the estimated $B^{-1}(t)$ graphical models across 90 brain regions as a function of age. Shown are results using three different values of the regularization parameter $\lambda$, and from left to right the kernel bandwidth parameter used is $h = 1.5$, 2, and 3 for both methods. Note the consistently increasing edge density in our estimate, corresponding to predictions of increased brain connectivity as the brain develops, and the difference in teenage development rates between healthy and ADHD patients.

(a) Histogram of the ages of all subjects in the ADHD-200 dataset.

(b) Histogram of the ages of healthy subjects in the ADHD-200 dataset.

(c) Histogram of the ages of subjects diagnosed with ADHD in the ADHD-200 dataset.

Figure 9: Histograms showing the age distributions of all patients, healthy patients, and ADHD diagnosed patients in the ADHD-200 dataset. A set of subjects in the dataset have no diagnosis, these left out of both the healthy and ADHD groups.

## 11 Additional experiments using alternate graph topologies

### 11.1 $A$ star-block and MA

In Figure 11, we repeat the experiments of main text Figure 2, showing results for $A$ changed to a star-block graph (edge weights defined as for ER), and $A$ an moving average (MA) covariance (band width 15). The results confirm the trends found for the AR case. Similarly, in Figure 12, we show the results for the $A$ estimator when $A$ is a star-block graph.

### 11.2 $B(t)$ random grid graph

In this section, we use a random grid graph which is produced in the same way as the random ER graph, except edges are only allowed on between adjacent nodes in a square 2 dimensional grid

Figure 10: Comparison of our method with KronPCA and the sample covariance. Shown is a logarithmic plot of the Frobenius norm error as a function of available replicates, with data generated using $A^{-1} \in \mathbf{R}^{60 \times 60}$, $B(t)^{-1} \in \mathbf{R}^{20 \times 20}$ random ER graphs.

Figure 11: MCC, Frobenius, and L2 norm error curves for $B$ a random ER graph and $n = 100$. From top to bottom: $A$ is star-block covariance, and MA covariance.

(Figure 13). We replace the random ER model for $B(t)$ used in the main text with the random grid graph model, and repeat the main text experiments using this alternate $B(t)$ topology.

Random grid graph results for the experiments shown in main text Figure 2 are shown in Figure 14, showing similar results as expected. Similarly, random grid graph results for the experiments shown in Figure 11 are shown in Figure 15.

For the $A$ part, Figure 16 repeats the experiments of main text Figure 3, and Figure 17 repeats the experiments of Figure 12, both using the random grid graph for $B(t)$.

## 12 Estimation error bound for $B$ part: Proof of Theorem 3

### 12.1 Preliminary results

Define $\tilde{B}(t_0)$ to be the expected value of the kernel-smoothed estimator $\hat{S}_m(t_0)$ at time $t_0$:

$$\tilde{B}(t_0) = E[\hat{S}_m(t_0)] = \sum_{i=1}^{m} w_t(t_0) B(i/m). \tag{31}$$

Figure 12: MCC, Frobenius, and L2 norm error curves for $A$ a Star-Block graph when $B$ is a random ER graph. From top to bottom: $m = 400$, $m = 800$, and $m = 1600$.

Using this notation, the estimator bias can be bounded via

**Lemma 5** (Bias). *Suppose there exists $C > 0$ such that $\max_{i,j} \sup_t |B''_{i,j}(t)| \leq C$. Then for a kernel $K(\cdot)$ satisfying assumptions (K1-K5) we have*

$$\sup_{t \in [0,1]} \max_{i,j} |\tilde{B}(t_0, i, j) - B(t_0, i, j)| = O\left(h + \frac{1}{m^2 h^5}\right).$$

This lemma is proved in Section 15.

The variance of the estimator $\hat{S}_m(t_0)$ can be bounded as:

**Lemma 6** (Variance). *Suppose $mh > n$. Define event $\mathcal{A}$*

$$\max_{i,j} |\hat{S}_m(t_0, i, j) - \tilde{B}(t_0, i, j)| \leq C\sqrt{\frac{\log mh}{mh}}, \tag{32}$$

*for some $C > 0$. Then $\mathbb{P}(\mathcal{A}) \geq 1 - \frac{c}{m^4 h^4}$.*

The proof of this result is based on an application of the Hanson-Wright inequality, and is found in the supplementary material. We emphasize that this bound holds for both the diagonal and off-diagonal elements simultaneously. Using a similar approach, in Section 12.6 we can also show that the estimator $\hat{S}_m(t_0)$ is positive definite with high probability:

**Lemma 7** (Positive definiteness). *Suppose $mh > n$. Define the event $\mathcal{B}$*

$$(1 + \delta)\tilde{B}(t_0) \succ \hat{S}_m(t_0) \succ (1 - \delta)\tilde{B}(t_0) \succ 0. \tag{33}$$

*for some fixed $\sqrt{\frac{c \log mh}{mh}} \leq \delta < 1$. Then $\mathbb{P}(\mathcal{B}) \geq 1 - \frac{c}{m^4 h^4}$.*

Figure 13: Example sequence of $B^{-1}(t) = \Theta(t)$ random grid graphs used in the experiments. At each time point, the 50 edges connecting $n = 100$ nodes are shown. Changes are indicated by red and green edges: red edges indicate edges that will be deleted in the next increment and green indicates new edges.

Figure 14: MCC, Frobenius, and L2 norm error curves for $B$ a random grid graph and $n = 100$. From top to bottom: $A$ is AR covariance with $\rho = .5$, AR covariance with $\rho = .95$.

## 12.2 Proof of Theorem 3

In this section, we derive the elementwise bound on the estimator $\widehat{S}_m(t_0)$ of the spatial covariance $B(t_0)$ at time $t_0$, and show that it is positive definite with high probability. To obtain the elementwise bound, we will first bound the bias and variance of $\widehat{S}_m(t_0, i, j)$ and then combine the bounds.

Figure 15: MCC, Frobenius, and L2 norm error curves for $B$ a random grid graph and $n = 100$. From top to bottom: $A$ is star-block covariance, and MA covariance.

Figure 16: MCC, Frobenius, and L2 norm error curves for $A$ a AR(1) with $\rho = 0.5$ when $B$ is a random grid graph. From top to bottom: $m = 200$ and $m = 800$.

## 12.3 Estimator bias bound

Recall that $\tilde{B}(t_0) = E[\hat{S}_m(t_0)]$. By Lemma 5 with $p = 1$ (proof in Section 15), we have

$$\sup_{t_0} \max_{i,j} |\tilde{B}(t_0, i, j) - B(t_0, i, j)| = O\left(h + \frac{1}{m^2 h^5}\right).$$

Figure 17: MCC, Frobenius, and L2 norm error curves for $A$ a Star-Block graph when $B$ is a random grid graph. From top to bottom: $m = 200$ and $m = 800$.

## 12.4  Estimator variance bound (Lemma 2)

**Lemma 2** (Variance). *Suppose $mh > n$. Define event $\mathcal{A}$*

$$\max_{i,j} |\hat{S}_m(t_0, i, j) - \tilde{B}(t_0, i, j)| \leq C\sqrt{\frac{\log mh}{mh}}, \tag{34}$$

*for some $C > 0$. Then*

$$\mathbb{P}(\mathcal{A}) \geq 1 - \frac{c}{m^4 h^4}. \tag{35}$$

*Proof.* Recall that

$$\widehat{S}_m(t_0) := \sum_{i=1}^{m} w_i(t_0) \left( X_i X_i^T - \frac{\text{tr}(A)}{m} I_n \right). \tag{36}$$

and

$$w_i(t_0) = \frac{1}{mh} K\left( \frac{i/m - t_0}{h} \right).$$

Then

$$\widehat{S}_m(t_0) - \tilde{B}(t_0) = \sum_{\ell=1}^{m} w_\ell(t_0)(X_\ell X_\ell^T - E X_\ell X_\ell^T).$$

Let $\hat{S} = \text{vec}(X)\text{vec}(X)^T \in \mathbb{R}^{mn \times mn}$ be the overall sample covariance. Then observe that for fixed $i, j, t_0$ there exists a vector $w^{ij}(t_0)$ with $\sum_t w_t^{(ij)}(t_0) = 1$ and $\|w^{(ij)}(t_0)\|_2 \leq c/\sqrt{mh}$ such that

$$\widehat{S}_m(t_0, i, j) - \tilde{B}(t_0, i, j) = [w^{(ij)}(t_0)]^T \text{vec}(\hat{S} - \Sigma).$$

By the triangle inequality, $\|\Sigma\| \leq \|A\| + \max_t \|B_t\|$, so $\|\Sigma\|\|w^{(ij)}(t_0)\|_2 \leq O(1/\sqrt{mh})$. We can thus apply Lemma 7 in Section 14, giving for fixed $i, j$

$$\mathbb{P}(|\widehat{S}_m(t_0, i, j) - \tilde{B}(t_0, i, j)| \geq \epsilon \|\Sigma\|\|w^{(ij)}(t_0)\|_2) = \mathbb{P}\left( [w^{(ij)}(t_0)]^T \text{vec}(\hat{S} - \Sigma) \geq \epsilon C \sqrt{\frac{1}{mh}} \right)$$

$$\leq 2 \exp\left( -c \frac{\epsilon^2}{K^4} \right).$$

Using the union bound over $i, j$ (cardinality $n^2$), the concentration bound Lemma 7 implies

$$\mathbb{P}\left(\max_{i,j} |\widehat{S}_m(t_0, i, j) - \tilde{B}(t_0, i, j)| \geq \epsilon C\sqrt{\frac{1}{mh}}\right) \leq n^2 2\exp\left(-c\frac{\epsilon^2}{K^4}\right)$$

$$= 2\exp\left(2\log n - c\frac{\epsilon^2}{K^4}\right).$$

Setting $\epsilon = c'\sqrt{\log mh}$, for large enough $c$ we have $2\log n - c\frac{\epsilon^2}{K^4} \leq 2\log n - c\log mh/K^4 \leq -c''\frac{\log mh}{K^4}$ since $mh > n$ and that

$$\max_{i,j} |\widehat{S}_m(t_0, i, j) - \tilde{B}(t_0, i, j)| \leq C\sqrt{\frac{\log mh}{mh}} \tag{37}$$

with probability at least $1 - \frac{c}{m^4 h^4}$.

$\square$

## 12.5 Total error

Putting the bias and variance together, we can bound the total error of the estimator. By the triangle inequality,

$$|\widehat{S}_m(t_0, i, j) - B(t_0, i, j)| \leq |\widehat{S}_m(t_0, i, j) - \tilde{B}(t_0, i, j)| + |\tilde{B}(t_0, i, j) - B(t_0, i, j)|.$$

Hence,

$$\max_{i,j} |\widehat{S}_m(t_0, i, j) - B(t_0, i, j)|$$

$$= O_p\left(h + \frac{1}{m^2 h^5} + \sqrt{\frac{\log m}{mh}}\right).$$

Optimizing over the order of $h$, we set $h \asymp \left(\frac{\log m}{m}\right)^{1/3}$, giving

$$\max_{i,j} |\widehat{S}_m(t_0, i, j) - B(t_0, i, j)| \leq C\left(\frac{\log m}{m}\right)^{1/3}. \tag{38}$$

for some $C$.

This completes the bound on the estimator error of $\widehat{S}_m(t_0)$. It remains to show that $\widehat{S}_m(t_0)$ is positive definite with high probability.

## 12.6 Positive definiteness of $\widehat{S}_m(t_0)$ (Proof of Lemma 7)

*Proof.* Let $u \in S^{n-1}$. Then

$$u^T \hat{S}_m(t_0) u = \text{vec}^T(uu^T)\text{vec}(\hat{S}_m(t_0))$$

$$= \text{vec}^T(uu^T)\left[\, w^{(1,1)}, \ldots, w^{(n,n)} \,\right]^T \text{vec}\left(\hat{S} - \frac{\text{tr}(A)}{m}I\right).$$

Observe that $\|\text{vec}^T(uu^T)\left[\, w^{(1,1)}, \ldots, w^{(n,n)} \,\right]^T\|_2 \leq c/\sqrt{mh}$, since the $w^{(ij)}$ have disjoint support. Thus by Lemma 7

$$\mathbb{P}\left(u^T(\hat{S}_m(t_0) - E[\hat{S}_m(t_0)])u > \epsilon\sqrt{\frac{c}{mh}}\right) \leq 2\exp(-C\epsilon^2) \tag{39}$$

Recall that $E[\hat{S}_m(t_0)] = \tilde{B}(t_0)$.

Then by a standard argument using the union bound over an $\epsilon$ net of $S^{n-1}$, which has cardinality $\leq \exp(n\log(3/\epsilon))$,

$$\mathbb{P}\left(\exists u \in S^{n-1} | u^T\left(S_m(t_0) - \tilde{B}\right)u \leq c\epsilon\right)$$

$$\leq \exp(n\log(3/\epsilon))2\exp(-C\epsilon^2 mh) \tag{40}$$

$$\leq C\exp\left(-c'\epsilon^2 mh\right)$$

This holds for $c$ large enough, since $n < mh$.

Suppose that the event in (40) holds. Then for all $u \in S^{n-1}$,

$$u^T S_m(t_0) u \geq u^T \tilde{B}(t_0) u - c\epsilon$$
$$\geq u^T \tilde{B}(t_0) u (1 - \delta)$$

where $\delta \lambda_{min}(\tilde{B}(t_0)) \geq c\epsilon$. Note that since $\tilde{B}(t_0)$ is a positively-weighted average of matrices $B_i$ with minimum eigenvalues $\geq 1/c_b$ (assumption A4), $\lambda_{min}(\tilde{B}(t_0)) \geq 1/c_b$. By a similar argument, the upper bound holds. We thus have

$$(1 + \delta)\tilde{B}(t_0) \succeq \hat{S}_m(t_0) \succeq (1 - \delta)\tilde{B}(t_0) \tag{41}$$

with probability at least $1 - \frac{c'}{m^4 h^4}$, for some fixed $\sqrt{\frac{c \log mh}{mh}} \leq \delta \leq 1$. Note that when (41) holds, $S_m(t_0)$ is positive definite. $\qquad \square$

## 12.7 Theorem 3

By the union bound the probability that the events $\mathcal{A}$ and $\mathcal{B}$ hold is $\mathbb{P}(\mathcal{A} \cap \mathcal{B}) = 1 - \frac{c}{m^4 h^4}$. Thus, combining the bound (38) and the proof of positive definiteness in the previous subsection, the proof of Theorem 3 in the main text results.

**Theorem 3.** *Suppose that the conditions of Lemma 2 hold and $h \asymp \left(\frac{\log m}{m}\right)^{1/3}$. Then with probability at least $1 - \frac{c''}{m^{8/3}}$,*

$$\max_{ij} |\hat{S}_m(t_0, i, j) - B(t_0, i, j)| \leq C \left(\frac{\log m}{m}\right)^{1/3}$$

*for some $C$, and $\hat{S}_m(t_0, i, j)$ is positive semidefinite.*

# 13 Estimation error bound for $A$ part: Proof of Theorem 6

## 13.1 Trace Estimator

We first bound the error for the estimator

$$\hat{tr}(B) = \sum_{i=1}^{m} w_i \|X_t\|_2^2 - \frac{n}{m} tr(A), \qquad w_i = \frac{1}{m}. \tag{42}$$

of the constant trace of $B$, $tr(B)$.

**Lemma 4.** *Suppose that $\|A\| \leq c_A$ and $\|B(t)\| \leq c_B$ for all $t, m$, and $tr(B(t))$ is constant over time. We have with probability $1 - \frac{3}{m^5}$,*

$$\frac{1}{n} |\hat{tr}(B) - tr(B)| \leq C(c_A + c_B) \sqrt{\frac{\log m}{mn}}.$$

*Proof.* The bias of $\hat{tr}(B(t_0))$ is zero since $tr(B(t))$ is constant. To bound the variance, we can rewrite (42) as

$$\hat{tr}(B(t)) = \|XW_t\|_F^2 - \frac{n}{m} tr(A),$$
$$W_t = \text{diag}(w).$$

Note that

$$E\|XW_t\|_F^2 = \sum_{i=1}^{m} w_i tr(B) + \frac{n}{m} tr(A) \tag{43}$$
$$= \tilde{tr}(B) + \frac{n}{m} tr(A). \tag{44}$$

Also note that $\|XW_t\|_F^2 = \mathrm{tr}(\mathrm{vec}(X)(W_t \otimes I_n)\mathrm{vec}^T(X))$. Thus,

$$\frac{1}{n}\hat{\mathrm{tr}}(B(t)) = \frac{1}{n}\mathrm{tr}\left(\mathrm{vec}(X)(W_t \otimes I_n)\mathrm{vec}^T(X)\right) - \frac{1}{m}\mathrm{tr}(A).$$

Hence, by Lemma 7, with $\tilde{w} = \frac{1}{n}(w(t) \otimes 1_n)$ on the $mn$ indices corresponding to the diagonal elements of $\hat{S} = \mathrm{vec}(X)\mathrm{vec}(X)^T$, we have for $\frac{\epsilon}{\sqrt{mn}} = o(1)$,

$$\mathbb{P}\left(|\tilde{w}^T\mathrm{vec}(\hat{S} - \Sigma)| \geq \epsilon(c_A + c_B)\|\tilde{w}\|_2\right) \leq 2\exp\left(-c\frac{\epsilon^2}{K^4}\right)$$

and thus

$$\mathbb{P}\left(\left|\frac{1}{n}(\hat{\mathrm{tr}}(B) - \mathrm{tr}(B))\right| \geq \epsilon(c_A + c_B)\frac{1}{\sqrt{mn}}\right) \leq 2\exp\left(-c\frac{\epsilon^2}{K^4}\right)$$

since $\|w\|_2 \leq \frac{1}{\sqrt{mn}}$. Set $\epsilon = C\sqrt{\log m}$ with $C$ such that with probability at least $1 - \frac{3}{m^5}$,

$$\frac{1}{n}|\hat{\mathrm{tr}}(B(t)) - \mathrm{tr}(B(t))| \leq C(c_A + c_B)\sqrt{\frac{\log m}{mn}}.$$

This concludes the proof. $\qquad\square$

## 13.2 Elementwise error (Proof of Theorem 6)

We can now prove the error bound for $\tilde{A}$.

**Theorem 6.** *Suppose Assumptions [B2, A1] hold. Then*

$$\max_{i,j}|\tilde{A}_{ij} - A_{ij}| \leq C(c_A + c_B)\sqrt{\frac{\log m}{n}}$$

*with probability* $1 - \frac{c}{m^4}$ *for some constants* $C, c > 0$.

*Proof.* Recall that

$$\tilde{A} = \frac{1}{n}X^TX - \frac{1}{n}\mathrm{diag}\{\hat{\mathrm{tr}}(B(1/m)), \ldots, \hat{\mathrm{tr}}(B(1))\}. \tag{45}$$

Recall that $\hat{S} = \mathrm{vec}(X)\mathrm{vec}(X)^T$. For $i \neq j$, by (45) and the definition of Kronecker products we can then write

$$\tilde{A}_{ij} = \frac{1}{n}X_i^T X_j$$

$$= \frac{1}{n}\sum_{\ell=1}^n \hat{S}_{\ell+(i-1)m, \ell+(j-1)m}$$

where $X_i$ is the $i$th column of $X$. Thus, we can write

$$\tilde{A}_{ij} = w^T\mathrm{vec}(\hat{S})$$

for some $w \in \mathbb{R}^{m^2n^2}$ with $n$ nonzero elements all equal to $1/n$. Clearly $\|w\|_2 = 1/\sqrt{n}$. We can then apply Lemma 7 with $w$ as the weight vector. Using the union bound over $i, j$ and assuming $\|A\|, \|B(t)\|$ are bounded by $c_a, c_b$ respectively, Lemma 7 thus gives

$$\max_{i \neq j}|\tilde{A}_{ij} - A_{ij}| \leq C(c_A + c_B)\sqrt{\frac{\log m}{n}} \tag{46}$$

with probability at least $1 - \frac{c}{m^4}$ for absolute constants $C, c$.

For the diagonal elements, Lemma 4 shows that with probability $1 - \frac{3}{m^5}$,

$$\frac{1}{n}|\hat{\mathrm{tr}}(B(t)) - \mathrm{tr}(B(t))| \leq C(c_A + c_B)\left(\frac{\log m}{mn}\right)^{1/2},$$

and thus by Lemma 7 and the union bound

$$\max_{i=j} |\tilde{A}_{ij} - A_{ij}| \leq C(c_A + c_B)\sqrt{\frac{\log m}{n}} + C_1(c_A + c_B)\left(\frac{\log m}{mn}\right)^{1/2}$$

$$\leq C(c_A + c_B)\sqrt{\frac{\log m}{n}}$$

with probability at least $1 - \frac{3}{m^4} - \frac{c}{m^4} > 1 - \frac{c}{m^4}$, since $m > n$.

Using the union bound gives

$$\max_{i,j} |\tilde{A}_{ij} - A_{ij}| \leq C(c_A + c_B)\sqrt{\frac{\log m}{n}} \tag{47}$$

with probability at least $1 - \frac{c}{m^4}$. $\qquad\square$

# 14  Concentration bound (Lemma 7)

We use the following concentration bound to bound the error of the $A$ and $B(t)$ estimates. Note that it also gives the corresponding $A$ part and $B$ part bounds in (Rudelson & Zhou, 2017) as special cases.

**Lemma 7** (Concentration bound). *Let $w \in \mathbb{R}^{m^2 n^2}$ and let $x = \Sigma^{1/2} z$ be a subgaussian random vector where $z_i$ are independent, zero mean, unit variance, and have $\|z_i\|_{\psi_2} \leq K$. Let $\hat{S} = xx^T$. Then if $\epsilon\|\Sigma\|\|w\|_2 = o(1)$*

$$\mathbb{P}(|w^T \mathrm{vec}(\hat{S} - \Sigma)| \geq \epsilon\|\Sigma\|\|w\|_2) \leq 2\exp\left(-c\frac{\epsilon^2}{K^4}\right) \tag{48}$$

*where $c$ is an absolute constant.*

*Proof.* By the definition of the vectorization operator and letting $\mathbf{x}_t(i) = [x_{t,(i-1)\bar{p}_1+1}, \ldots, x_{t,i\bar{p}_1}]$,

$$w^T \mathrm{vec}(\hat{S}) = w^T \mathrm{vec}(xx^T) = x^T W x = z^T \Sigma^{1/2} W \Sigma^{1/2} z, \tag{49}$$

where $W = \mathrm{vec}^{-1}(w)$ and $\mathrm{vec}^{-1}(\cdot)$ is the inverse of the vectorization operator, mapping $\mathbb{R}^{m^2 n^2}$ to $\mathbb{R}^{mn \times mn}$.

Thus, by the Hanson-Wright inequality,

$$\mathbb{P}(|w^T\mathrm{vec}(\hat{S} - \Sigma)| \geq t) \leq 2\exp\left(-c\min\left(\frac{t^2}{K^4\|\Sigma^{1/2}W\Sigma^{1/2}\|_F^2}, \frac{t}{K^2\|\Sigma^{1/2}W\Sigma^{1/2}\|}\right)\right)$$

$$\leq 2\exp\left(-c\min\left(\frac{t^2}{K^4\|\Sigma\|^2\|W\|_F^2}, \frac{t}{K^2\|\Sigma\|\|W\|}\right)\right)$$

$$\leq 2\exp\left(-c\min\left(\frac{t^2}{K^4\|\Sigma\|^2\|w\|_2^2}, \frac{t}{K^2\|\Sigma\|\|w\|_2}\right)\right)$$

so

$$\mathbb{P}(|w^T\mathrm{vec}(\hat{S} - \Sigma)| \geq \epsilon\|\Sigma\|\|w\|_2) \leq 2\exp\left(-c\frac{\epsilon^2}{K^4}\right)$$

where we set $t = \epsilon\|\Sigma\|\|w\|_2$ and assumed $\epsilon\|\Sigma\|\|w\|_2 = o(1)$.

$\qquad\square$

# 15  Kernel Smoothing (Proof of Lemma 5)

The following lemma bounds the bias inherent in using a kernel to smooth the sample covariance matrix for $B(t)$ across time.

***Proof of Lemma 5***. The proof is found in Lemma 5 of (Zhou et al., 2010), repeated here for completeness.

Without loss of generality, let $t = t_0$. We will use the Riemann integral to approximate the sum

$$\tilde{B}_p(t_0, i, j) = \frac{1}{m} \sum_{k=1}^{m} \frac{2}{h} K\left(\frac{\frac{k}{m} - t_0}{h}\right) B_{i,j}\left(\frac{k}{m}\right).$$

Specifically,

$$\tilde{B}_p(t_0, i, j) = \int_0^1 \frac{2}{h} K\left(\frac{u - t_0}{h}\right) B_{i,j}(u) du + O\left(\frac{1}{m^2 h^5}\right)$$

where the second equality follows from Assumption K5 and the assumed bound on the second derivative of $B_{ij}(t)$. From the proof of Lemma 5 in (Zhou et al., 2010), $\int_0^1 \frac{2}{h} K(\frac{u-t_0}{h}) B_{i,j}(u) du - B_{i,j}(t_0) = O(h)$ so

$$\tilde{B}_p(t_0, i, j) - B_{i,j}(t_0) = O\left(h + \frac{1}{m^2 h^5}\right).$$

Taking the maximum over $i, j$ and $t_0 \in [0, 1]$ completes the proof.

$\square$