[Reviews · NeurIPS 2017]

Reviewer 1



This paper presents an elegant theoretical framework that enables estimating a covariance on both time and features. The framework is based on decomposing the covariance in two additive terms, a temporal and a spatial one, with temporal variations. It seems that the authors uploaded a second copy of the manuscript as the supplementary materials. This is a pity. The projection performed in eq (10) seems ad hoc; necessary undoubtedly, but I do not see what it corresponds to in terms of statistical model. While the model is interesting, the validation on real data is a bit convoluted. With regards to fMRI, could the authors give insight on the structure of B^-1 estimated? I am not sure that the general Markov model for the time domain is useful. The validation is very indirect: a increase in brain connectivity is interesting, but it would be of more neuroscientific interest to quantify where it happens or what is its nature. Also, it is unclear to me why this is related to estimating time-changing connectivity. It is a pity that the authors chose to work on the ADHD 200 data, which is one of most noisy large resting-state dataset. Little solid results have come out of this dataset. To study development, I would have trusted more the Philadelphia Neurodevelopmental Cohort. Sparse inverse covariance techniques have been used in fMRI since at least 2010. I find it surprising that the authors are citing only papers from 2015 and after. The fonts are tiny on figure 2 and 3. They are very hard to read. The conclusion mentions validation on two fMRI data, but I see only one: ADHD 200. When the authors write \|A\|_max, I believe that they have in mind the max norm, often written \|A\|_\infty. It would have been good to make this explicite.

Reviewer 2



This paper studied the graphical structure in spatiotemporal data through transferring the time series data into an additive model, by assuming stationary temporal correlation structure and time-varying undirected Gaussian graphical model for spatial correlation structure. With the assumption that the spatial correlations change smoothly with time, they proposed estimators for both spatial and temporal structures based on kernel method and GLasso approach. The statistical convergence property of the estimators was provided under certain assumptions. The approach presented good performance in both simulation and fMRI data application studies. This paper is overall clearly written, with solid theoretical support and interesting application. However, some assumptions, such as a common temporal correlation matrix for different spatial locations, may be not valid in application. It may worth further discussion. There are also some other points may need further improvement. 1. The authors didn’t provide correct supplementary, but another version of the main text. 2. In the estimation section, tr(A) is supposed to be known. This is not so obviously satisfied in real data application. It is better to provide more discussions on this issue. 3. Line 178 is unclear. Which lemma? It should be the inverse of B, instead of B itself, exploiting sparsity. 4. As m / n increases, the bound in Theorem 2 / Theorem 4 converges. However, the regularization parameter \lambda_m / \lambda_n also converges to 0 at the same time, which is a sacrifice of sparsity. In simulation, the authors used a variety of regularization parameters, instead of the proposed one in the theorem. More discussions are expected on this issue. 5. In simulation, it is recommended to compare with other methods, such as Qiu (2016). Also the authors mentioned that some previous methods were limited to AR(1) model, while their method in this paper can handle more complex models. But they only provide simulation results for AR(1) case, as shown in Fig. 2 & 3. So it is expected to see the simulation results on more general models. 6. In simulation, it would be better to use the inverse of B(t) instead of B(t) for evaluation, as the focus is the structure of the underlying undirected graphs instead of the edge weights / strength of the undirected graphs. 7. The assumption that all spatial points follow the same temporal structure is a bit strong. In application, its validity should be discussed. For example, in the fMRI data, different brain regions may present different developing pattern. The visual cortex may develop fast in childhood, while the dorsal prefrontal cortex related to decision and attention usually has a development peak in a later age. It would be better to check the validity of these assumptions before application. 8. Assuming a time series follows a joint normal distribution (equation (2)) is also not very common. In the fMRI data application, its validity should also be discussed. Each of the 90 regions of interest generally has hundreds to thousands voxels, which should enable the test for normality. 9. In Line 118, it assumes that spatial structure is sparse and m much larger than n. However, the brain connectivity is usually not sparse. In the meantime, in this resting-state fMRI dataset, some subjects has less time points (m larger than or equal to 76) than the number of ROIs (n = 90). 10. In the application, the dataset include both healthy group and ADHD group. Did the authors use them all together for analysis? It might be better to apply the method to each group separately and compare the results. Different developing trends are expected. And if such a result is obtained, it should be more powerful to illustrate the effectiveness of this method. 11. In consistency statements for ‘signal’ and ‘noise’, in sense of their correspondence with ‘temporal’ and ‘spatial’ structure. In abstract (line 3-4), they mentioned ‘temporally correlated signal’ and ‘spatially correlated noise’. However, in 2.1 estimator section (line 133 – 134), they mentioned ‘spatial signal variance’ and ‘noise variance … corresponding to tr(A)’, while A corresponds to temporal structure as indicated in equation (2). 12. In the paragraph from line 44 to 51, it is better to add comparison with Qiu (2016) and Monti (2014), in sense of methodology. 13. Some minor points: (i) It is better to provide the definition of e_i after equation (4), as it is provided after equation (5). (ii) The subscripts in equation (8) are not consistent, where ‘t’ should be ‘i’. (iii) In line 170, B to B(t). (iv) In line196, m to n. Update: I am satisfied with the authors' replies to my comments. Hence I would like to raise my grade.

Reviewer 3



This work considers a time-varying Gaussian graphical model for spatio-temporal data. A nonparametric method is used to estimate both time-scale and spatial-scale covariance matrix. My comments are as follows: 1. In equations (2) and (3), the normality assumption is a strong one since the spatio-temporal data may not follow multivariate normal in both rows and columns. 2. The notation in this work is somehow confusing. The X is sometimes denoted as random variable, and sometimes denoted as random matrix. 3. One major concern is that the estimation procedure of B relies on the estimation of A, while the estimation of A also replies on the estimation of B. But the author does not use the iterative procedure to enable the close loop. Also it is not clear how is the convergence of such an iterative procedure. Another concern is that there are quite a few tuning parameters in the proposed method. It is not clear how the authors choose the optimal values of the multiple tuning parameters. 4. In this simulation study, the authors only show the performance of the proposed method in terms of the tuning parameter. But it is unclear what is the performance of the proposed method with the optimally-chosen lambda. 5. The real application is not very convincing to show the advantage of the proposed method.